# Microbiological Diagnoses on Clinical Mastitis—Comparison between Diagnoses Made in Veterinary Clinics versus in Laboratory Applying MALDI-TOF MS

**DOI:** 10.3390/antibiotics11020271

**Published:** 2022-02-19

**Authors:** Lærke Boye Astrup, Karl Pedersen, Michael Farre

**Affiliations:** 1Center for Diagnostics, Technical University of Denmark, Kemitorvet, 2800 Kgs Lyngby, Denmark; 2Department of Animal Health and Antimicrobial Strategies, National Veterinary Institute, 751 89 Uppsala, Sweden; karl.pedersen@sva.se; 3SEGES Livestock Innovation, Agro Food Park 15, 8200 Aarhus, Denmark; mifa@seges.dk

**Keywords:** mastitis, MALDI-TOF MS, major pathogens, diagnostics, antibiotics

## Abstract

The present study compares the diagnoses on clinical bovine mastitis made in veterinary clinics using conventional diagnostic methods with diagnoses on the same samples made by a veterinary reference laboratory using MALDI-TOF MS as diagnostics. The study enables targeted and evidence-based consulting on prudent mastitis diagnostics and related antibiotic usage. In total, 492 samples from clinical mastitis were included. When applying MALDI-TOF MS as gold standard, only 90 out of 492 diagnoses made in veterinary clinics, equal to 18%, were correct. Four main findings were important: (1) the veterinary clinics overlooked contamination in mastitis samples; (2) the veterinary clinics only assigned 2 fully correct diagnoses out of 119 samples with mixed growth cultures; (3) the veterinary clinics made close to half of their diagnoses on pure culture erroneously; (4) the veterinary clinics applied a limited number of the relevant pathogen identifications on pure culture samples. Altogether, the present study shows that a large part of Danish clinical mastitis cases are misdiagnosed. Lack of correct diagnoses and diagnostic quality control may lead to the choice of wrong treatment and thus hamper prudent use of antibiotics. Hence, the present study warns a risk of overuse of antibiotics in Denmark. Consequently, the present study calls for training of veterinary clinics in diagnostics of mastitis pathogens and national guidelines on quality assurance of mastitis diagnostics.

## 1. Introduction

Mastitis is one of the most frequent and costly diseases in dairy cattle [1,2]. Mastitis also accounts for a large part of antibiotic consumption in adult dairy cattle [3,4] and compromises animal welfare [5]. Therefore, numerous strategies have been developed both on a national and international level to decrease the occurrence of bovine mastitis and its negative effects [2,6,7]. However, no matter the strategies, mastitis prevention and treatment depend on accurate microbiological diagnoses.

In mastitis, certain bacteria are considered major as they are frequently reported as causes of mastitis and/or causes of mastitis-related economical loss including increased culling or mortality [8,9,10]. *Escherichia coli*, *Klebsiella* spp., *Staphylococcus aureus*, *Streptococcus uberis*, *Streptococcus dysgalactiae* and *Streptococcus agalactiae* are considered as major pathogens [10]. However, it is well established that bacterial prevalences in mastitis varies both within and between countries, and over time [6,11,12]. In Denmark, the estimated prevalences of mastitis bacteria relies on recordings of the diagnoses made by private veterinary clinics. Since the early 1980s, most Danish veterinary clinics obtain their diagnoses based on inhouse microbiological analysis. These analyses are performed without any formal quality assurance or accreditation. Most often, the applied analyses include Gram-stain or potassium hydroxide test, catalase test, coagulase test, microscopy and morphological characterization based on both blood agar and CHROMagar^TM^ Orientation or other selective and indicative agars. Such analyses results in a limited set of phenotypic biochemical characteristics and do not appropriately differentiate the variety of species that can be present in milk and will grow on blood agar. Hence, despite being laborious, such analyses only allow for tentative identification of a limited range of bacteria—mainly the major pathogens. These limitations have become more and more evident with the gains of molecular techniques, disguising the extensive variety of mastitis-causing pathogens [7,13,14,15,16]. The present study aimed to evaluate the veterinary clinic inhouse mastitis diagnoses by confirmation testing with matrix-assisted laser desorption–ionization time-of-flight mass spectrometry (MALDI-TOF MS) identifications as gold standard. This comparison provides the ostensible accuracy of (1) the decision support for antibiotic treatment in each single mastitis case and (2) the collective Danish register data on mastitis diagnoses. Thereby, the study enables targeted and evidence-based consulting of Danish veterinarians on prudent diagnostics and related antibiotic usage. Moreover, by providing an example of a national mastitis approach based on biochemical analyses, the study contributes to the ongoing international debate on how to ensure evidence-based mastitis diagnoses and treatments in the era of molecular diagnostics.

## 2. Results

In total, 492 agar plates (hereafter, samples) were shipped to the Technical University of Denmark (DTU) and examined by Good Laboratory Practice microbiological quality control and MALDI-TOF MS in line with National Mastitis Council (NMC) guidelines [17]. Out of these 492 samples, 158 were positive for more than two pathogen species (i.e., deemed contaminated). In addition, 68 samples were culture negative on arrival at DTU. Accordingly, 266 samples were included in the study. Out of 266 samples, 147 contained a pure culture and 119 contained two bacterial species (mixed growth status samples, hereafter just “mixed growth samples” [12]). The status of the samples is shown in Table 1.

To evaluate the diagnostic accuracy of the diagnoses assigned by the veterinary clinics, the results were analyzed on two levels: status and pathogen identification, respectively. Together the status and the pathogen identification(s) make up the diagnosis. Results are presented for pure-culture samples and mixed-growth samples separately in the following.

Results on pure culture samples: The MALDI-TOF MS analysis identified 147 pure cultures out of the 492 samples provided by the veterinary clinics. The 147 MALDI-TOF MS diagnoses were compared to the diagnoses (status plus pathogen identification(s)) assigned in the submission form from the veterinary clinics. The status “pure culture” was correctly assigned to 132 of the 147 samples by the veterinary clinics. The remaining 15 samples were assigned no pathogen identification and erroneous status by the veterinary clinics (one sample was assigned status “contaminated”, three samples assigned status “culture negative”, and 11 samples assigned neither status nor pathogen identification). Based on the accepted pooling of pathogens, the MALDI-TOF MS confirmed the pathogen identification assigned by the veterinary clinics in 77 out of the 132 pure cultures. Thus, in the 147 pure cultures in total, the assigned diagnosis was erroneous in 70 samples, out of which 55 erroneous diagnoses were due to erroneous pathogen identification and 15 erroneous diagnoses were due to lack of pathogen identification and erroneous status. In the 132 pure culture samples, in which the veterinary clinics did assign a pathogen identification, only 9 different pathogen identifications were used by the veterinary clinics in total. Results are shown in Table 2. In contrast, the MALDI-TOF MS analysis of the pure culture samples identified 26 different pathogens. Results shown in Table 3.

When considering pathogen identification on species level, the veterinary clinics assigned only 5 pathogen identifications correctly on pure-culture samples (*S. aureus*, *S. uberis*, *E. coli*, *S. dysgalactiae* and *T. pyogenes*) out of 26 species assigned by MALDI-TOF MS (Table 2 and Table 3). These 5 pathogen identifications were correctly assigned in 55 out of 147 pure culture ientifications in total (Table 2). When applying the grouping of pathogens, the MALDI-TOF MS identification comprised 17 different pathogen identifications (Table 2). Despite allowing for grouping of pathogens, eight different diagnoses from eight different genera were never applied by the veterinary clinics (as shown in Table 2, and as marked by an asterix in Table 3). Among these, the diagnoses *S. agalactiae* and *C. bovis*/*Corynebacterium* spp. were never applied by veterinary clinics. From a total of 147 cases identified as clinical mastitis with a pure-culture milk sample, major pathogens accounted for 100 (68%) of the samples (30 *S. aureus*, 27 *S. uberis*, 26 *E. coli*, 9 *Klebsiella* spp., 7 *S. dysgalactiae* and 1 *S. agalactiae*) (Table 3). Collectively, the veterinary clinics assigned the correct status of pure culture to 132 out of 147 samples. Out of the 132 samples assigned correct status, 77 samples were assigned a correct pathogen identification too, and hence a correct diagnosis. Out of the remaining 70 pure cultures, 55 were assigned a correct status but an erroneous pathogen identification. The 55 samples with correct status but erroneous pathogen identification contained a similar distribution of pathogens as the correctly identified 77 samples, both when regarding the pathogen identifications assigned by veterinary clinics and those assigned by MALDI-TOF MS, respectively. The 55 samples included 39 pure cultures of major pathogens (results not shown but can be calculated by subtracting the number of major pathogens correctly identified by veterinary clinics from the number of major pathogens identified by MALDI-TOF MS in Table 2). Likewise, the 77 samples assigned correct status and correct pathogen identification included 61 pure cultures of major pathogens (24 *S. aureus*, 24 *E. coli*, 7 *Klebsiella* spp., 4 *S. uberis*, 2 *S. dysgalactiae* and 0 *S. agalactiae*, Table 2). Moreover, the 15 samples assigned an erroneous status, including the 11 samples assigned status “no pathogen identification” by the veterinary clinics, showed no particularly different distribution of pathogens than the other pure cultures, when analyzed by MALDI-TOF MS (results not shown).

Results on mixed growth status samples: The MALDI-TOF MS analysis identified 119 samples with mixed growth. Out of these, the veterinary clinics assigned the correct status (i.e., two pathogen identifications) to four samples, two of which had only one or zero pathogen identifications confirmed, respectively. Hence, 2 out of 119 mixed growth samples were diagnosed fully correctly by the veterinary clinics. Out of the remaining 115 samples with erroneous status, 16 were assigned neither status nor pathogen identification by the veterinary clinics. The last 99 mixed growth samples (115 − 16) were erroneously assigned a pure culture status by the veterinary clinics. Of these 99 samples, 31 samples had mixed growth consisting of two different NAS. Due to the accepted pooling of the NAS-group, a pure-culture diagnosis of NAS for these samples were considered correct. This was the case for 11 out of the 31 samples. Another 24 out of the 99 samples that were erroneously assigned a pure-culture status, had a non-NAS pathogen correctly identified by the veterinary clinics. Accordingly, 64 samples (115 − (11 + 24)) with mixed growth were assigned both erroneous status and erroneous pathogen identification by the veterinary clinics.

The samples with mixed growth that triggered an erroneous diagnosis were composed of the same type of pathogens, as the samples that had one or both pathogens identified correctly. As such, there seemed to be no pattern in the erroneous diagnoses. As an example, the two samples with mixed growth that were correctly diagnosed by the veterinary clinics both contained a combination of *E. coli* and *S. aureus*. However, the same combination, along with other combinations of two major pathogens, had only 1 or 0 pathogens correctly identified in 19 and 5 samples, respectively (in total, the mixed samples included 26 samples composed of 2 major pathogens, results not shown).

Collectively, the results on mixed-growth samples showed that the status “mixed growth” was assigned correctly in four samples and overlooked or neglected in 115 samples. The pathogens present in the mixed-growth samples were the same species as those idenifyed in the pure culture samples, except that the pathogens *Lactococcus lactis*, *Pantoea agglomerans*, *Providencia heimbachae*, *Serratia Liquefaciens*, *Streptococcus gallolyticus*, *Streptococcus parauberis* and *Yersinia enterocolitica* were found in mixed-growth samples only. These seven different pathogens were found in one mixed-growth sample each.

Summing up the different types of erroneous diagnoses from veterinary clinics, 90 out of 492 diagnoses, equal to 18%, were correct (77 diagnoses on pure cultures, 2 fully correct diagnoses on mixed growth samples, and 11 diagnoses on mixed growth samples considered correct when grouping the NAS, respectively).

## 3. Discussion

Our study showed a considerable discrepancy between diagnoses assigned based on quality control and MALDI-TOF MS compared to diagnoses assigned in veterinary clinics based on colony morphology and biochemical characteristics. In particular, four main findings were important: (1) the veterinary clinics overlooked or neglected contamination in 158 out of 492 mastitis samples in total; (2) the veterinary clinics only assigned 2 fully correct diagnoses on 119 samples with mixed-growth cultures; (3) the veterinary clinics made close to half of their diagnoses on pure cultures erroneously (70 out of 147, 48%), even when accepting a considerable grouping of pathogens; (4) the veterinary clinics applied a much-limited number of the relevant pathogen identifications on pure-culture samples—in total, diagnoses from veterinary clinics on pure-culture samples reflected only 9 out of 17 groups of pathogens (Table 2) and only 5 out of 26 exact species diagnoses.

Based on the four main findings, we strongly suggest national guidelines on quality assurance of mastitis diagnostics in veterinary practice. Additionally, we strongly suggest that the individual veterinary practitioner reconsider the merits of continuing their inhouse diagnostic service. Biochemical tests are laborious and only offer a limited possibility to differentiate the variety of relevant mastitis-causing pathogens, unless a very wide range of tests are applied, which was not the case in any of the involved practices. Hence, biochemical tests might neither provide the optimum decision support nor the most cost-effective methods to treat and prevent mastitis. However, detailed considerations of pros and cons for inhouse diagnostics based on biochemical analyses lies without the scope of this text. We therefore refer veterinary practitioners to consult the literature on the matter of choosing the most beneficial diagnostic approaches, including replacing or supplying biochemical inhouse diagnostics with out-lab services [7,15,16]. Targeted training courses for staff performing microbiological analyses in veterinary clinics must be recommended as well.

The results in the present study indicate that less than every fifth case of clinical mastitis is diagnosed adequately and correctly in Denmark. Dividing the results analysis into “status” and “pathogen identifications” illustrates that erroneous status of contaminated and mixed-growth samples drives many erroneous diagnoses (158 and 64 out of 492 samples in the present study, respectively). In the present study it is not possible to determine whether the veterinary clinics overlooked the contamination and the mixed growth, or neglected them. Regardless of why, a surprisingly large part of samples were assigned a wrong status, despite the specific and elaborate instructions that were given on discharge of contaminated samples at the beginning of the study. Hence, it seems that the eagerness of identifying a mastitis pathogen vastly overrules the objective and thorough analysis of the samples, including assigning a correct status to them. Neglecting the sample status is problematic, even if an important mastitis pathogen is present. First of all, it is not possible to assess whether if the sample is truly mixed growth or contaminated, if only one pathogen out of multiple is analyzed/identified. In many of the samples that turned out to be contaminated in the present study, the spreading of milk on the fourstar plate provided by the veterinary clinics was too dense, which makes it difficult to visually recognize or even suspect the presence of contamination. Lack of quality-assured sample preparation including subculturing when necessary thus seems part of the explanation of the many erroneous diagnoses. Secondly, if the sample is truly mixed growth it is not possible to conclude which pathogen is the important one out of the two—or if both should be considered a possible cause of the mastitis case, if the sample is erroneously considered a pure culture. To draw such conclusions, it is necessary to identify both species present, along with the approximate quantity of both. As examples, 24 of the samples identified as a pure culture by the veterinary clinics proved to contain two major pathogens such as *E. coli* and *S. uberis*. Others that were assigned a pure-culture status by the veterinary clinics proved to contain a fast and a more slow growing pathogen, such as corynebacteria, when incubated for 48 h. These examples illustrate why it is unjustified to identify one of the present pathogens only. Last but not least, according to the NMC, contaminated samples should be graded before any further decisions can be made on the sample [17]. Contaminated samples can be graded as either “gross contamination” or “low-level contamination” with the former grade leading to discharge of the sample, and with the latter grade including identification of (at least) the dominant pathogen. In conclusion, assigning only one pathogen identification to a sample with more pathogens present makes it impossible to conclude whether the sample is mixed or contaminated, and to conclude what pathogen is the most probable cause of disease.

Regarding the pathogen identifications, the present study shows that many samples are erroneously diagnosed, even when the sample is a pure culture and a major pathogen is present. Moreover, the present study shows that many different pathogens can be present in samples from clinical mastitis, as it has been shown in many former studies [11,13,18]. The bacteria identified most often in the present study consists of those considered major pathogens, which is in line with international expectations for clinical mastitis [6,10,11]. However, when summing up the bacteria usually considered as minor/other mastitis pathogens, these were in fact detected in almost one third of the pure culture samples (47 cases out of 147, 32%). Most notably, the Non-Aureus Staphylococci (NAS) as a group was present in many pure-culture samples (21 samples out of 147). This underlines the need to not only focus on major pathogens in clinical mastitis, and hence the need for veterinary clinics to perform more detailed diagnostics in order to ensure evidence-based consulting and prudent use of antibiotics. The need for detailed diagnostics is even more important when taking into account that minor pathogens, the NAS in particular, might be even more often present in mixed growth samples and in subclinical mastitis [10,19,20,21] and thereby be underestimated in the present study. Moreover, the large number of different pathogen species in clinical mastitis calls for attention in the increasing use of PCR as a diagnostic tool in mastitis diagnostics. If applied in general, mastitis diagnostics including PCR kits, need to be able to detect many different pathogens in milk samples [11,13,18], which is not the case for the PCR kits currently available in Denmark. If relevant PCR primers are lacking and/or only genus-specific primers are available, there is a considerable risk of false-negative or oversimplified PCR diagnoses. Such false-negative diagnoses pose an obvious problem to the individual mastitis case as consulting and/or treatment may be misguided. Furthermore, if veterinarians solve their diagnostic in-house challenges by replacing culture with oversimplified PCR-kits—or any other analysis that is carried out in an oversimplified way, this might reinforce the risk of disfiguring the collective register data on mastitis etiology. Therefore, our study highlights the need for quality assurance of inhouse diagnostics as well as of out-lab services.

Many studies already point out that correct diagnoses are a prerequisite for proper treatment of mastitis. Accordingly, the need for more accurate diagnostic tools for point-of-care use has been established for years, as has the need for further development and proper use of molecular diagnostic tools [15,16,22]. The present study supports these needs.

One important limitation to the present study is that the collective MALDI-TOF MS diagnoses identified cannot be considered true prevalences. This is because the samples provided to DTU might have been preselected despite the fact that veterinary clinics were asked to provide all but their contaminated plates from clinical mastitis cases. Nevertheless, our study indicates a considerable variation of mastitis pathogens, which is in accordance with other studies [11,13,18]. Another limitation is the relatively low number of samples included in our study. Yet, it can be counterargued that the 11 participating veterinary clinics represent a large part of the Danish veterinarians engaged in mastitis diagnostics. Moreover, in our research group, another study is currently being conducted with more milk samples from cows with mastitis and more farms included. The preliminary results of the ongoing study seems to confirm the diagnostic problems described in the present study. Lastly, an argument that might be held against the present study is that only a limited range of antibiotics are available for mastitis treatment in Denmark. Thus, several different diagnoses might eventually lead to the same type of treatment. However, an important part of mastitis treatment ought to be evidence-based deselection of treatment. As such, correct diagnoses are a prerequisite in prudent use of antibiotics as it enables both selecting and deselecting treatment. Likewise, herd specific consulting on mastitis prevention requires correct diagnoses too.

Altogether, the present study underlines the need for detailed diagnostics in mastitis treatment and consulting. Detailed diagnostics are necessary to ensure correct diagnoses which are of special importance for three reasons: first, targeted consultancy and mastitis prevention cannot be achieved without correct diagnoses; second, treatment choices including treatment-deselection cannot be targeted without correct diagnoses; third, proper national data quality on mastitis cannot be acquired without correct diagnoses. No matter how much data is registered, the prevalences will always reflect the diagnoses we are able to perform. Consequently, as long as diagnostic inadequacy prevails, we will never know the distribution and causes of mastitis nationally, and we will never secure evidence-based consulting and treatment (de-)selection in the singular case.

## 4. Materials and Methods

A total of 11 veterinary clinics specialized within dairy cattle participated in the study. The veterinarians provided milk cultures along with their own diagnoses to the National Veterinary Institute, Technical University of Denmark (Now Center for Diagnostic DTU). Samples were collected from 31 dairy farms from October throughout December 2016. Sampling included only quarter milk samples from clinical mastitis cases (all levels of clinical mastitis were accepted, the only inclusion criteria was presence of clinical signs and no current treatment, thus subclinical mastitis was not included). The veterinary clinics were provided with Selma Plus^®^ four-star agar plates (National Veterinary Institute, Uppsala, Sweden). The four parts of each plate were: Bovine blood agar, MacConkey agar, Mannitol salt agar, and PGUA agar, respectively (Section of Substrate Production, National Veterinary Institute, Sweden). The supply of agar plates aimed to ensure that all samples were provided in the same manner as DTU and that no extra costs were imposed on the veterinarians, thereby avoiding risk of selection bias. Before sampling, a one-day seminar was held for the veterinarians to ensure compliance with the project protocol. The veterinarians then conducted their own diagnostic routines as usual and only discarded contaminated samples (defined as samples with >2 different pathogen species on the agar plate [17]). After fulfilling the in-house diagnostics, the veterinarians shipped the samples to DTU. At DTU, all submitted plates were visually inspected and quality controlled. Damaged plates were discarded. Contaminated plates defined by growth of three or more colony types were not further investigated. From plates with apparent growth of one or two colony types, a representative colony of each type was subcultured on blood agar (5% calf blood, SSI Diagnostica A/S, Hillerød, Denmark) and incubated at 37 °C and read after 24 and 48 h. All subcultures that seemed pure cultures were subjected to identification to species level by using MALDI-TOF MS (Bruker Daltonics, Bremen, Germany). Colonies were identified by direct deposition on the target plate according to Bizzini et al. [23], combined with overlay of 70% formic acid prior to matrix deposition using the BDAL database combined with the DTU-Vet database for veterinary spectra and staphyloccoci [24,25].

To evaluate the diagnostic accuracy of the diagnoses assigned by the veterinary clinics, the results were analyzed on two levels: status and pathogen identification, respectively. Together, the status and the pathogen identification(s) make up the diagnosis. Hence, the MALDI-TOF MS results were used to conclud (1) whether if the status “pure culture” and “mixed growth” was assigned correctly by the veterinary clinics or not and (2) whether the particular pathogen identification(s) assigned to pure cultures and mixed-growth samples by the veterinary clinics was correct or not. That way, the present study takes into consideration that a pathogen can be correctly identified despite the fact that the sample is assigned a wrong status and vice versa.

Samples that proved contaminated based on MALDI-TOF MS identification on subcultures were excluded from the results. However, Non-Aureus Staphylococci (NAS) were considered as a group rather than as individual species in the definition of contamination. The MALDI-TOF MS pathogen identifications on both mixed-growth samples (2 species) and pure-culture samples were informed to the veterinary clinics. The MALDI-TOF MS pathogen identifications were compared to the veterinary diagnoses (status plus pathogen identification(s)). The comparison was based on acceptance of certain bacterial species being pooled by the veterinarians. The accepted pooling of pathogens were: the group “NAS” rather than the exact species names of all Non-Aureus Staphylococci, the genera *Micrococcus* spp.”, “*Enterococcus* spp.”, “*Lactococcus* spp.”, “*Bacillus* spp.”, “*Corynebacterium* spp.”, “*Pseudomonas* spp.”, “*Pasturella* spp.”, “*Klebsiella* ssp.”, “*Citrobacter* spp.”, *Proteus* spp., “*Enterobacter* spp.” and “*Serratia* spp.“ rather than exact species from these genera, the identification “other streptococci” for all streptococci other than *S. uberis*, *S. agalactiae* and *S. dysgalactiae*, and the pathogen identifications “yeast” and “algae” rather than the exact genus and possibly species names of all yeasts and algae. In total, the veterinary clinics provided 492 Selma Plus agar plates to DTU.

## 5. Conclusions

According to this study, the majority of Danish clinical mastitis cases are misdiagnosed. The erroneous diagnoses are caused by the neglect of three important diagnostic aspects: mixed growth infections, contaminated samples, and the variation within mastitis causes. Neglecting these aspects, veterinary clinics not only misdiagnose the singular mastitis case, but also overlook much of the variation in total among mastitis causing pathogens, thereby disfiguring the national collective data on mastitis. Hence the present study highlights a considerable need for quality assurance of mastitis diagnostics in veterinary practice in order to ensure prudent use of antibiotics in Denmark.

## Figures and Tables

**Table 1 antibiotics-11-00271-t001:** Status of the total number of 492 agar plates provided to DTU by the veterinary clinics.

Status of Sample Based on Quality Control and MALDI-TOF MS	Number of Samples Provided by 11 Veterinary Clinics	Number of Correct Status on Sample Assigned by Veterinary Clinic
Pure culture	147	132 ^1^
Mixed growth samples (2 species)	119	4 ^2^
Contaminated samples (>2 species)	158	0 ^3^
Culture-negative samples	68	0 ^4^
Total	492	136

^1^ Veterinary clinics erroneously assigned the status “Contaminated”, “Culture negative”, or “No diagnosis” to viable pure cultures in 1, 3 and 11 samples respectively. Accordingly, out of 147 pure cultures, 132 were assigned a correct status (i.e., a single pathogen identification) by the veterinary clinics. ^2–4^ All samples that were found by quality control and MALDI-TOF MS to be either contaminated or culture negative were erroneously assigned a single-pathogen (i.e., pure-culture) diagnosis by the providing veterinary clinics. All but 4 out of 119 samples that were found by quality control and MALDI-TOF MS to be mixed growth, were erroneously assigned a single-pathogen diagnosis by the providing veterinary clinics as well. Hence, the veterinary clinics assigned erroneous sample status to all but four samples, except the pure cultures.

**Table 2 antibiotics-11-00271-t002:** Comparative pathogen identifications with pathogen-groupings, assigned by MALDI-TOF MS and by veterinary clinics, respectively, on pure-culture clinical mastitis samples plated on Selma Plus^®^ four-star agar.

Pathogen Identification with Pathogen Groupings	Number of Pathogen Identifications by MALDI-TOF MS	Number of Pathogen Identifications by Veterinary Clinics (Number of MALDI-OF MS Confirmed Pathogen Identifications)
*Staphylococcus aureus*	30	28 (24)
*Streptococcus uberis*	27	9 (4)
*Escherichia coli*	26	31 (24)
Non-*aureus* staphylococci	21	26 (14)
*Klebsiella* spp.	9	8 (7)
Yeast	7	2 (1)
*Streptococcus dysgalactiae*	7	18 (2)
*Trueperella pyogenes*	6	2 (1)
*Corynebacterium bovis*	5	0 (0)
*Enterococcus saccharolyticus*	2	0 (0)
Other streptococci	2	8 (0)
*Acinetobacter* spp.	1	0 (0)
*Aerococcus viridans*	1	0 (0)
*Bacillus simplex*	1	0 (0)
*Proteus vulgaris*	1	0 (0)
*Streptococcus agalactiae*	1	0 (0)
*Magnusiomyces capitalis*	1	0 (0)
Contaminated	0	1 (0)
Culture-negative	0	3 (0)
No diagnosis	0	11 (0)
Total	147	147 (77)

For each pathogen, the table shows the number of true positives, according to MALDI-TOF MS, along with the true positive, false positive and false negative identifications assigned by the veterinary clinics. As example, 30 samples out of 147 were true positive for *S. aureus*. The veterinary clinics assigned the pathogen identification “*S. aureus*” 28 times to the 147 pure cultures in total. Of these 28, the veterinary clinics identified 24 correctly. Hence the veterinary clinics made 4 (28 − 24) false positive *S. aureus* diagnoses and 6 (30 − 24) false negative *S. aureus* diagnoses in 147 pure cultures, respectively, and so forth.

**Table 3 antibiotics-11-00271-t003:** MALDI-TOF MS pathogen identifications without pathogen-groupings, in the total number of 147 samples with pure cultures provided to DTU by the veterinary clinics.

Major vs. Minor/Other Pathogens	MALDI-TOF MS Pathogen Identification without Pathogen Groupings	Number of Samples
Major pathogens	*Staphylococcus aureus*	30
	*Streptococcus uberis*	27
	*Escherichia coli*	26
	*Klebsiella pneumoniae*	8
	*Streptococcus dysgalactiae*	7
	*Klebsiella oxytoca*	1
	*Streptococcus agalactiae* *	1
	**Major pathogens in total**	**100**
Minor/other pathogens	*Staphylococcus simulans*	9
	*Trueperella pyogenes*	6
	*Corynebacterium bovis* *	5
	*Staphylococcus chromogenes*	5
	*Candida tropicalis*	4
	*Staphylococcus epidermidis*	3
	*Enterococcus saccharolyticus* *	2
	*Staphylococcus haemolyticus*	2
	*Aerococcus viridans**	2
	*Acinetobacter* spp. *	1
	*Bacillus simplex* *	1
	*Candida krusei*	1
	*Candida rugosa*	1
	*Magnusiomyces capitalis **	1
	*Proteus vulgaris* *	1
	*Staphylococcus arlettae*	1
	*Staphylococcus sciuri*	1
	*Streptococcus gallolyticus*	1
	**Minor/other pathogens in total**	**47**
	**All pathogens in total**	**147**

* Denotes pathogen identifications that were never applied on pure-culture samples by the veterinary clinics.

## Data Availability

The datasets generated and analysed during this study are available upon reasonable request.

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
