# Peer review of "Microbiological Diagnoses on Clinical Mastitis—Comparison between Diagnoses Made in Veterinary Clinics versus in Laboratory Applying MALDI-TOF MS"

_antibiotics, 2022, doi:10.3390/antibiotics11020271_

Round 1

Reviewer 1 Report

The manuscript entitled " Comparison of Microbiological Diagnoses on Acute Clinical 2 Mastitis made by Veterinary Clinics versus MALDI-TOF MS" represents a considerable amount of work. The following comments need to be addressed before the manuscript is suitable for publication in antibiotics Journal.

  • Line 58: please change evidencebased to evidence-based and also change diagnostes to diagnoses
  • Line 235; please change erroneusly to erroneously.
  • Why didn't authors statically analyze their results?
  • Please, prove your paper with more and new references
  • I think this manuscript should be short communication

Reviewer 2 Report

Despite moderate editing of the English language is required, the manuscript is well organized. The information presented is new and interesting and the conclusions are supported by the data.

Therefore, I have only some minor revisions to suggest, as detailed below:

  • I would suggest showing the mixed bacterial infection in the table.
  • I did not understand if the isolates to be molecularly characterized were really randomized or if they were made based on the results of the microbiological analysis.

Reviewer 3 Report

The manuscript is entirely out of scope for this journal.
There is not a single word in the text about antibiotics and treatment of mastitis.
If the editors have agreed that beforehand with the authors (for obvious reasons ....), then the reviewers should have been notified in advance about the situation.
Moreover, there is very little novelty in the diagnostics part of the study. MALDI-TOF approaches are now commercially available and are being used routinely for the diagnosis of mastitis pathogens, hence, there is very little scope to publish this study.

In all, the manuscript merits rejection, but it is appreciated that the editors of the special issue might for some obvious reasons support this for publication, hence I return a major revision opinion to allow the manuscript to advance in the next stage.

Reviewer 4 Report

Dear Authors,

in my opinion the manuscript covers very important topic regarding mastits diagnostics in cattle. Generally, the study design and methodology is correct, however the manuscript would benefit from language improvement. I found the title as well as abstract not clear. Please rewrite it.

Please find below my main questions and doubts.

  1. Are you sure that obtained milk (discharge) samples were collected only from clinical mastitis samples ? If yes, what were the visual abnormalities in milk. Were the cows presented with any systemic symptoms ? I ask because many of S. aureus cases remain subclinical.
  2. Why you wrote "acute" in the title ? 
  3. Please change the legends below Table 2. You can directly in the table indicate, which pathogens are considered as minor by giving correct heading.
    Major pathogens

                  S. aureus....

                    E. coli...

         Minor pathogens

           Candida...

4. Just a slight - suggestion - you can draw a simple graph indicating milk samples proceeding ( how many you received, how many and because of what reason where excluded, how many eventually were examined). I am a bit suprised by the number of excluded samples. 

Minor suggestions:

Line 31 - change to compromises or compromised

Line 41-42 - Escherichia coli, Streptococcus... are considered as major pathogens.

line 46- 1980s

line 58 - evidence-based

line 89 - erroneous

line 124 - biochemical characteristics

line 134 - suggest

line 149 - particular

line 155 - in milk samples

line 172 - milk samples from cows with mastitis

line 183 - of special importance

line 184 - targeted -> achieved

lines 187 - 194 - please rewrite it to "reported speech" - do not use pronoun (we)

lines 234 - diagnosed erroneusly -> misdiagnosed

line 239 - warn -> highlights 

There are discrepancies between tables and discussion in nomenclature. Please modify and unify this (CNS, non-aureus staphylococci etc.)

Reference number 17 is incomplete - please check it

Round 2

Reviewer 3 Report

The authors have greatly improved the manuscript, which now is of quality acceptable for publication. This has become an excellent piece of work that certainly merits publication and sets a fantastic example of novelty in the topic of mastitis treatment.
Subject to improving a few linguistic slips after minor revision, the corrected manuscript is fully suitable for publications.